# Investigation of Physical Characteristics and In Vitro Anti-Inflammatory Effects of Fucoidan from *Padina arborescens*: A Comprehensive Assessment against Lipopolysaccharide-Induced Inflammation

**DOI:** 10.3390/md22030109

**Published:** 2024-02-27

**Authors:** Hyo-Geun Lee, N. M. Liyanage, Fengqi Yang, Young-Sang Kim, Seung-Hong Lee, Seok-Chun Ko, Hye-Won Yang, You-Jin Jeon

**Affiliations:** 1Department of Marine Life Sciences, Jeju National University, Jeju 63243, Republic of Korea; hyogeunlee92@jejunu.ac.kr (H.-G.L.); liyanagenm1992@jejunu.ac.kr (N.M.L.); yfq426@naver.com (F.Y.); medieval032@gmail.com (Y.-S.K.); 2Marine Science Institute, Jeju National University, Jeju 63333, Republic of Korea; 3Department of Pharmaceutical Engineering, Soonchunhyang University, Asan 31538, Republic of Korea; seunghong0815@gmail.com; 4National Marine Biodiversity Institute of Korea, 75, Jangsan-ro 101-gil, Janghang-eup, Seocheon 33362, Republic of Korea; seokchunk@mabik.re.kr

**Keywords:** *Padina arborescens*, fucoidan, anti-inflammatory activity, structural characteristics, functional food

## Abstract

A biocompatible, heterogeneous, fucose-rich, sulfated polysaccharide (fucoidan) is biosynthesized in brown seaweed. In this study, fucoidan was isolated from *Padina arborescens* (PAC) using celluclast-assisted extraction, purified, and evaluated for its anti-inflammatory potential in lipopolysaccharide (LPS)-induced RAW 264.7 cells. Structural analyses were performed using Fourier transform infrared (FTIR) and scanning electron microscopy. Among the purified fucoidans, fucoidan fraction 5 (F5) exhibited strong inhibitory activity against LPS-induced nitric oxide (NO) production and pro-inflammatory cytokine generation through the regulation of iNOS/COX-2, MAPK, and NF-κB signaling in LPS-induced RAW 264.7 cells. Determination of the structural characteristics indicated that purified F5 exhibited characteristics similar to those of commercial fucoidan. In addition, further analyses suggested that F5 inhibits LPS-induced toxicity, cell death, and NO generation in zebrafish models. Taken together, these findings imply that *P. arborescens* fucoidans have exceptional anti-inflammatory action, both in vitro and in vivo, and that they may have prospective uses in the functional food sector.

## 1. Introduction

Inflammation is a common immunological defense mechanism involving the physiological and pathological processes of the immune system. Inflammatory stimulants induce macrophage activation and increase the production of inflammatory mediators, such as nitric oxide (NO) and prostaglandins, pro-inflammatory cytokines (IL-1, IL-6, TNF-α), and inducible nitric oxide synthase (iNOS) and cyclooxygenase-2 (COX-2). [1]. However, uncontrolled and excessive inflammatory responses lead to tissue damage, pathogenesis of immunological diseases, and increased proliferation of cancer cells [2]. Earlier publications revealed that this could increase the incidence of chronic inflammatory disorders, such as atherosclerosis, interstitial cystitis, dermatitis, multiple sclerosis, and bowel syndrome [3]. Owing to their safety and limited adverse effects, natural substances for the treatment of such disorders have recently gained popularity. Natural substances that inhibit the synthesis of inflammatory molecules can be used as medicines or anti-inflammatory agents.

Seaweeds have received considerable attention and popularity owing to their newly discovered secondary metabolites with various biological functionalities [4]. Recently, there has been an increasing trend towards the application of seaweed-derived bioactive compounds in the food and functional food industries [5]. Sulfated polysaccharides from brown seaweed are renowned for various biological activities, such as antioxidant, anti-inflammatory, and cosmeceutical activities [6]. Most earlier studies have reported that the sulfated polysaccharide fucoidan from brown seaweeds possesses multifunctional, non-toxic, and highly biocompatible polymers [7]. Fucoidans from different marine algae have many multifunctional bioactivities, including anti-inflammatory, antitumor, antioxidant, pro-immune, and antiviral effects [8,9,10,11]. Enzymatic extraction is thought to be more advantageous than the traditional techniques for extracting bioactive compounds from seaweeds, including microwave-assisted extraction, ultrasound-assisted extraction, supercritical fluid extraction, and pressurized liquid extraction [12]. The biological activity of fucoidan has been identified based on its structural traits, such as its monosaccharide composition, sulfate content, and molecular weight. Its structure and content vary according to the type of brown alga from which it is extracted, the area, and the extraction method.

*Padina arborescens* (*P. arborescens*) is a species of brown macroalgae commonly known as the “tree-like *Padina*”. *P. arborescens* inhabits intertidal zones and grows in shallow waters. It is a well-known food ingredient and marine herb in Korea and Japan. Despite the potential of *P. arborescens* to be a sustainable source of natural bioactive compounds, its functional value remains largely unknown. At present, there are many publications on the active compounds of *P. arborescens*. One study reported the hypoglycemic effect of *P. arborescens* extract in streptozotocin-induced diabetic mice, whereas another study reported its antioxidant activity against glucose-induced apoptosis in pancreatic cells [13,14]. However, relatively few studies have been conducted on fucoidans isolated from *P. arborescens*. In the present study, we investigated the anti-inflammatory characteristics of fucoidans isolated from *P. arborescens* in South Korea. Our findings support the use of this compound in industrial biotechnological applications, including the manufacture of pharmaceuticals and functional foods, and offer new insights into its utilization.

## 2. Results

### 2.1. Composition of P. arborescens Extracts

Compositional analysis was performed according to the official methods of the Association of Official Analytical Chemists (AOAC) International. Celluclast-assisted extraction was used to extract fucoidan from *Padina arborescens* (PAC), exhibiting a higher yield (26.00 ± 0.71%) than that obtained using deionized water extraction (9.00 ± 1.53%). Figure 1 shows the diethylaminoethyl cellulose (DEAE-C) chromatogram of the crude polysaccharide obtained from PAC (PACC). The DEAE-C chromatogram indicated that five fucoidan fractions (F1–F5) were collected from PACC. Table 1 illustrates the chemical compositions of PAC, PACC, and the fucoidan fractions. Among the five fucoidan fractions, high contents of polysaccharide were observed in F1 (74.55 ± 1.01%). Chemical composition analysis revealed that the total polysaccharide content gradually decreased with successive fractions, whereas the sulfate content gradually increased with DEAE-C purification. The highest sulfate content was observed in F5 (38.74 ± 1.05%).

### 2.2. Polysaccharide Characterization through FTIR and Monosaccharide Profiling

Fourier transform infrared (FTIR) and monosaccharide analyses were performed on the purified fucoidan fractions to clarify their structural properties and functional groups. The wavelength range of 500–2000 cm^−1^ is frequently used to characterize the structural properties of fucoidans. As shown in Figure 2a, the FTIR spectrum of commercial fucoidan exhibited prominent peaks at 845, 1035, 1220–1270, and 1620 cm^−1^, respectively. A slight difference was observed for the minor peak at 845 cm^−1^ and the overlapped peak at 1035−1220 cm^−1^ in F5. The peak at 845 cm^−1^ is attributed to the bending vibration of the C-O-S bond, while the prominent peak at 1035 cm^−1^ is assigned to the C-O-C glycosidic bond. The broadened peak at 1220–1270 cm^−1^ represents the stretching vibration of the S=O bond in the sulfate groups. The signal at 1620 cm^−1^ indicates H-O-H bending vibration. According to the results, F5 exhibited a sharp and prominent peak at 1220–1270 cm^−1^, indicating that the degree of sulfation was increased in F5.

### 2.3. Morphology and Size of F5

The morphological characteristics and size differences of PAC, PACC, and F5 are shown in Figure 2b. Our findings indicated that the particles in PAC and PACC ranged in size from 50 to 200 µm and from 160 to 400 µm, respectively. F5 possessed particles less than 50 µm in size. Scanning electron microscope (SEM) images indicated that PAC and PACC exhibited varying surface particle sizes and relatively large, flattened, and rough surfaces compared to F5, which exhibited a regular, thin, thread-like structure.

### 2.4. Monosaccharide Composition

Figure 2c shows the monosaccharide compositions of PAC, PACC, and F5. A considerable increase in fucose and galactose was observed in F5 compared to PAC and PACC. After purification, the amounts of fucose and galactose increased, whereas the glucose content decreased.

### 2.5. F5 Showed the Highest Protective Activity among Five Fucoidan Fractions

The effects of the PAC, PACC, and fucoidan fractions (F1–F5) were examined for their impact on cytotoxicity and NO generation in RAW 264.7 cells. As shown in Figure 3a, at doses of 25, 50, 100, and 200 µg/mL, PAC, PACC, and F1–F5 treatment did not substantially reduce the viability of the cells, indicating low toxicity and no negative effects in RAW 264.7 cells. However, compared with unstimulated cells, lipopolysaccharide (LPS) stimulation of RAW 264.7 cells led to a discernible decline in cell viability and an increase in nitric oxide (NO) generation. Interestingly, cotreatment with fucoidan and LPS improved cell survival and negated the negative effects of LPS on macrophages (Figure 3b). Moreover, all fucoidan fractions reduced NO production in LPS-stimulated cells in a dose-dependent manner (Figure 3c). These findings led to the selection of F5 for further experiments because of its higher activity than the other fractions.

### 2.6. F5 Inhibits the Release of PGE2 and Inflammatory Cytokines from LPS-stimulated RAW 264.7 Cells

Pro-inflammatory cytokines and PGE2 are linked to apoptotic signaling and phagocytosis and play crucial roles in inflammation [15]. As a result, we investigated how F5 affects macrophages’ ability to secrete pro-inflammatory cytokines, such as tumor necrosis factor (TNF)-α, interleukin (IL)-6, and IL-1β, and PGE2. Pro-inflammatory cytokine and PGE2 release was significantly increased after LPS stimulation of RAW 264.7 cells, as shown in Figure 4, and this effect was inhibited in a concentration-dependent manner through pretreatment with F5.

### 2.7. F5 Inhibits the Production of iNOS and COX-2 in LPS-induced RAW 264.7 Cells

Western blotting was used to examine the protein expression levels and assess the effect of F5 on the expression of iNOS and COX-2 (Figure 5). The findings demonstrated that LPS stimulation increased the iNOS and COX-2 levels, which may account for the elevated NO and PGE2 expression in activated RAW 264.7 cells. Overall, the anti-inflammatory effect of F5 is supported by the fact that F5 treatment reduced the upregulation of iNOS and COX-2 in LPS-stimulated cells in a dose-dependent manner.

### 2.8. F5 downregulates the Over-Expression of NF-κB and MAPK Pathway Proteins

To evaluate the mechanisms underlying the effects of F5 on the iNOS, COX-2, and pro-inflammatory cytokine levels, we examined NF-κB and MAPK signaling pathway protein expression. The NF-κB and MAPK signaling pathways are essential for numerous cellular processes in eukaryotic cells. Inflammation may be reduced through NF-κB and MAPK phosphorylation inhibition [16]. As shown in Figure 6a, LPS treatment increased the NF-κB cascade, which led to the phosphorylation of p50/p65 and its translocation into the nucleus. Due to LPS stimulation, the phosphorylation of p65 and p50 increased in comparison to the control group. However, pre-incubation with F5 significantly decreased the phosphorylation of NF-κB-related proteins. The level of protein activation in the MAPK signaling pathway was measured 30 min after LPS stimulation to determine the effect of F5 on this pathway. LPS stimulation initiated the activation of the MAPK signaling pathway (Figure 6b). Significant phosphorylation of MAPK proteins, including p38 and JNK, was induced through LPS stimulation, and the band intensities of these proteins were greater than those in the unstimulated group. In conclusion, F5 demonstrated a strong ability to downregulate elevated NF-κB and MAPK protein expression levels.

### 2.9. F5 Protects Zebrafish Larvae from LPS-Induced Toxicity

The percentage of surviving zebrafish embryos was measured to assess the toxicity of F5 (Figure 7a). No toxicity was observed at concentrations of 50 and 100 µg/mL; however, 200 µg/mL of F5 was toxic to embryos. Therefore, the 50 and 100 µg/mL concentrations were used for further experiments. The survival of zebrafish larvae decreased drastically after LPS treatment (Figure 7b). Pre-incubation with F5 increased cell survival in a dose-dependent manner. Additionally, owing to stress activation, the heart rate of zebrafish treated with LPS was much higher than that of the control group (Figure 7c). However, LPS-induced stress was reduced after F5 treatment.

### 2.10. F5 Protects Zebrafish against LPS-induced Cell Death, NO, and ROS Production

As shown in Figure 8, LPS stimulation increased cell death and NO and reactive oxygen species (ROS) production in zebrafish. Based on these findings, the positive control group fluoresced strongly after stimulation with LPS, whereas F5 treatment significantly decreased the fluorescence intensity in a dose-dependent manner.

## 3. Discussion

The exploitation of marine organisms has become an important topic in industry. Brown algae are recognized as sustainable sources of bioactive secondary metabolites and nutritional food components [17]. This study provides an understanding of the physical characteristics of fucoidans isolated from *P. arborescens* and their potential anti-inflammatory activities.

To obtain functional ingredients, such as fucoidan, from seaweed, enzyme-assisted extraction (EME) is frequently used. Industrially, EME offers several advantages over other extraction methods, such as higher extraction efficiency, high yield, and low production cost. Several studies have reported that enzyme-assisted extracts exhibit non-toxicity and enhanced bioactivity [18]. Previous studies have reported that selective EME using specific enzymes results in differential bioactivities [18]. Celluclasts are a type of cellulase that catalyze the breakdown of glucose polymers in the cell wall structure [19]. In our previous study, celluclast-assisted extraction exhibited a high yield and remarkable antioxidant and anti-inflammatory activities [20]. In addition, a number of studies have investigated the anti-inflammatory effects of celluclast-assisted extracts from brown seaweed species [19,21]. The present study used celluclast EME to obtain crude sulfated polysaccharides from *P. arborescens*, and the obtained sample was further purified using DEAE column chromatography, yielding five fractions. Polysaccharides, proteins, polyphenols, and sulfates were present in each fraction, and the lower protein and polyphenol levels in each fraction demonstrated the efficiency of the anion exchange column technology.

Initial studies were conducted to identify the fucoidan fraction with the highest activity. F5 exhibited a greater inhibition of LPS-induced NO generation than the other fractions. Additionally, F5 had the highest sulfate concentration among the fractions, which is crucial for its anti-inflammatory effects [22]. Therefore, F5 was selected for further investigation. FTIR spectroscopy is necessary to determine the functional groups of polysaccharides, and it is based on the vibrational analysis of the functional groups in the structures [19]. Vibrational spectroscopic results of F5 revealed a similar pattern to commercial fucoidan at 845, 1035, and 1220–1270 cm^−1^. Furthermore, F5 exhibited an intensive peak at 1220–1270 cm^−1^, which confirmed that it possessed similar structural features to commercial fucoidan [19].

Monosaccharide composition results indicated an increase in the fucose and galactose levels in F5. These results suggest that fucose and galactose were highly concentrated during the DEAE purification step, and they are consistent with previous fucoidan purification studies on brown seaweed species [23]. According to a previously published study, the biological functions of ester sulfates in fucoidan contribute to cellular identification by specifically attaching to cell surface receptors [24]. Moreover, the antioxidant, anti-inflammatory, and immunomodulatory properties are greater in fucoidans with higher sulfate groups. In addition, purification modified the morphology of the particles, which was supported by simultaneous SEM examination, demonstrating a direct correlation between reduced particle size and increased bioactivity potential [25]. The SEM findings from these investigations indicated that a decrease in particle size enhanced anti-inflammatory efficacy.

Macrophages play a role in inflammation and become active when they encounter inflammatory stimuli, such as LPS. In the current investigation, the generation of inflammatory reactions and the activation of RAW 264.7 macrophages were assessed using LPS as a reference. LPS is composed of lipids and polysaccharides that are classified as bacterial toxins, and they are responsible for inducing inflammatory reactions in RAW 264.7, which are characterized by an increase in NO production. However, pathological conditions that cause excessive NO production may have negative consequences, such as cytotoxicity, resulting in tissue damage [20]. Exposure to LPS stimulates NO production and release by cytokine-activated macrophages through iNOS synthesis. Excessive production of NO causes cellular inflammation and tissue damage [26]. Therefore, inhibition of NO production is essential. In the present study, we established that F5 reduced LPS-induced NO generation. Our findings are consistent with those of earlier investigations on *Chnoospora-minima*- and *Sargassum-swartzii*-derived fucoidans [27,28].

Another inflammatory disease mediator, PGE2, is produced by cells in response to external stimuli. Excess PGE2 release results in detrimental inflammatory reactions and tissue damage. The pathophysiology of inflammation is influenced by immune cells, including neutrophils, macrophages, and eosinophils, that produce pro-inflammatory cytokines, such as TNF-α, IL-6, and IL-1β. LPS stimulation increased the production of NO, PGE2, and pro-inflammatory cytokines in the RAW 264.7 cells in the current study. These negative effects were ameliorated by F5 treatment. Similar outcomes were observed in studies on the anti-inflammatory effects of fucoidan extracted from *C. minimum*, *Ecklonia cava*, and *Saccharina japonica* on LPS-induced inflammation [19]. NO secretion plays a significant role in immunological modulation and is associated with iNOS and COX-2 [29]. Western blot analysis has shown that sulfated polysaccharides, such as fucoidan produced by seaweed, suppress iNOS and COX-2 in stimulated cells. Typically, during inflammation, cells with elevated iNOS levels have overexpressed COX-2. NO production is controlled by arginine oxidation via NOS, whereas COX-2 expression is associated with PGE2 synthesis [30]. We demonstrated that F5 treatment downregulated the LPS-induced upregulation of iNOS and COX-2 protein expression. Similar results were observed when *S. swartzii* was analyzed for its ability to reduce inflammation in LPS-induced RAW 264.7 cells [21]. However, F5 exhibited lower efficacy compared to the COX inhibitor indomethacin [31]. This suggests that the natural compound F5, while showing lower efficacy, is non-toxic, highlighting its potential use as a raw material for health functional foods. Collectively, these findings suggest that F5 treatment suppresses COX-2 and iNOS expression in LPS-induced RAW 264.7 cells.

The activation of pro-inflammatory mediators and cytokines, such as IL-1β and TNF-α, as well as the production of pro-inflammatory genes are mediated by the NF-κB signaling system, which is a significant signaling cascade. LPS induces NF-κB dimer phosphorylation and translocation to the nucleus, which, in turn, triggers the production of iNOS and COX-2 proteins [32]. In accordance with previous studies, we found that LPS stimulation enhanced the phosphorylation of p65 and p50, which was suppressed by F5 treatment in a dose-dependent manner. Numerous intricate molecular pathways are involved in inflammation. The MAPK signaling pathway plays a crucial role in controlling cytokines and pro-inflammatory mediators [33]. p38, one of the three significant MAPKs (JNK, ERK, and p38), regulates the production of inflammatory regulators. Furthermore, ERK, JNK, and p38 are important integrators in pharmacological therapy for inflammatory disorders. Multiple studies have shown that LPS can activate the MAPK signaling pathway by phosphorylating MAPK proteins [8]. Our results revealed that F5 downregulated the expression of MAPK pathway proteins in LPS-stimulated RAW 264.7 cells.

For further confirmation, in vivo inflammation-related experiments were performed. Due to the functional and structural similarities of the substrate enzyme-binding areas to their human equivalents, zebrafish are a useful model animal for studying a variety of diseases. Consequently, this served as an in vivo model for examining the antioxidant activity of F5. The ROS production levels measured in zebrafish indicated that F5 treatment resulted in declining intensities, revealing its protective effect against LPS-induced ROS generation. Moreover, cell death and NO generation are crucial processes for maintaining homeostasis. Tissues are damaged through the production of many intracellular cytokines, immune system activation, inflammation, excessive cell death, and NO production [34]. In this study, we found that co-treatment with F5 significantly decreased the LPS-induced elevation of NO and cell death.

## 4. Materials and Methods

### 4.1. Chemicals and Reagents

Food-grade celluclast was obtained from Novozyme (Novozyme Nordisk, Bagsvaerd, Denmark). Dulbecco’s Modified Eagle medium (DMEM), fetal bovine serum (FBS), penicillin–streptomycin (P/S), and trypsin-EDTA were purchased from Welgene, Inc. (Daegu, Korea). Gallic acid, glucose, Folin & Ciocalteu’s phenol reagent, peroxidase, hydrogen peroxide (H_2_O_2_), 3-(4,5-dimethylthiazol-2-yl)-2,5-diphenyltetrazolium bromide (MTT), gum arabic, and lipopolysaccharide (LPS) were purchased from Sigma-Aldrich (St. Louis, MO, USA). Dimethyl sulfoxide (VWR International, West Chester, PA, USA) and bovine serum albumin (Bovogen, VIC, Australia) were also used. The bicinchoninic acid (BCA) protein assay kit was purchased from Thermo Scientific (Waltham, MA, USA). Barium chloride dihydrate, ethanol, iron (II) sulfate heptahydrate, acetic acid, hydrochloric acid, nitric acid, ammonium sulfate, sodium hydroxide, phenol, and sulfuric acid were purchased from Dae Jung (Seoul, South Korea). The primary antibody NF-κB p65 (Catalog: 8242S), phospho NF-κB p65 (Catalog: 3033S), NF-κB p50 (Catalog: 3035S), MAPK p38 (Catalog: 8690L), phosphorylated MAPK p38 (Catalog: 9211S), and iNOS (Catalog: 2982) were purchased from Cell Signaling Technology (Danvers, MA, USA). The β-actin (Catalog: SC-47778), phospho NF-κB p50 (Catalog: SC-271908), and COX-2 (Catalog: SC-376861) were purchased from Satna Cruz Biotechnology (Dallas, TX, USA). All analytical reagents were purchased for use in the experiments. 

### 4.2. Sample Collection, Enzyme-Assisted Extraction, and Fucoidan Purification

*P. arborescens* was collected along the shores of Jeju Island, Korea (33.4602° N, 126.9353° E) in March 2019. Collected *P. arborescens* was completely washed with tap water to remove salt and sand, lyophilized, and ground into a powder, which was stored at −20 °C in a freezer. The resulting *P. arborescens* powder was subjected to EME. First, phenolic chemicals and pigments were removed from powdered *P. arborescens* through depigmentation with 10% formaldehyde under EtOH agitation for 4 h. The remaining formaldehyde was removed using ethanol and air-dried, and *P. arborescens* powder was recovered using enzyme-aided hydrolysis. Briefly, 4 g of fine *P. arborescens* powder and 40 mg of celluclast (1%) were suspended in 400 mL of deionized water. Thereafter, the mixtures were adjusted to be within optimal pH (4.50) and temperature (50 °C) and placed in a shaking incubator for 24 h. After extraction, enzymatic activity was stopped through heat inactivation. After heat inactivation, the mixtures were centrifuged and filtered. The collected supernatant was referred to as the celluclast-assisted extract from *Padina arborescens* (PAC). To obtain the crude polysaccharide (CPS), PAC was directly precipitated in 95% EtOH at 4 °C, and the precipitated crude polysaccharide (PACC) was recovered through centrifugation. The obtained CPS was dried and homogenized using a grinder. The resulting PACC was loaded onto diethylaminoethyl cellulose (DEAE-C) equilibrated with a 50 mM sodium acetate buffer. The loaded PACC was fractionated using a gradient of NaCl (0, 50, 100, 200, 400, 800, and 1000–2000 mM) at a flow rate of 2 mL/min. This resulted in five fucoidan fractions: F1–F5.

### 4.3. Chemical Composition Analysis of Purified Fucoidan Fractions 

The AOAC technique was used for chemical composition analysis [35]. The total polysaccharide content was measured through phenol–sulfuric acid analysis using glucose as the standard material [36]. The protein, polyphenol, and sulfate contents were measured using the Lowry method, the Folin–Ciocalteu method, and barium chloride analysis, respectively [37].

### 4.4. Selection of Fucoidan Fraction 

The murine macrophage cell line, RAW 264.7, was maintained in DMEM with recurrent subculturing. Cells exhibiting exponential growth were used for subsequent assays. NO production and cell viability of RAW 264.7 cells were tested for the five fucoidan fractions to determine which fucoidan fraction had the highest activity. The sample toxicity and NO generation were assessed using previously described techniques [20]. RAW 264.7 cells were seeded in 96-well plates at a concentration of 1 × 10^5^ cells/mL to test the sample toxicity and incubated for 24 h in a CO_2_ incubator. The cells were subsequently incubated with various doses of the fucoidan fractions (25, 50, 100, and 200 μg/mL). The MTT assay was performed following a 24 h incubation period. The Griess test was used to measure NO production. The seeded cells were treated with LPS (1 µg/mL with or without the extracts (25–200 µg/mL)) for 24 h. After incubation, an equivalent volume of the Griess reagent was added to the cell-free supernatant to measure the absorbance. F5 was selected for further experiments because of its outstanding activity.

### 4.5. Monosaccharide Compositions Analysis

Purified fucoidan was analyzed using a Dionex Bio-LC system (Dionex, Sunnyvale, CA, USA) and a high-performance anion exchange chromatography system equipped with a Dionex-ED 50 electrochemical detector (Thermo Fisher Scientific, Waltham, MA, USA). A CarboPac PA1 column (4.5 mm × 50 mm) was used for monosaccharide separation. Separation was performed via isocratic elution with 18 mM of NaCl at a flow rate of 1 mL/min. The detection waveform of the detector was set at 0.05 potential waveform. Standard mixtures (fucose, rhamnose, arabinose, galactose, glucose, xylose, and fructose) were injected as described above [8].

### 4.6. Evaluation of Surface Morphology

The surface morphologies of PAC, PACC, and F5 were determined using field-emission scanning electron microscopy (FE-SEM; MIRA 3 TESCAN, Czech Republic) equipped with energy-dispersive X-ray spectrometry (EDS). For morphological analysis, the samples were mounted onto a circular aluminum stub coated with carbon tape and scanned using FE-SEM under the setting conditions (magnification: ×300 and ×1000; accelerating voltage: 3.0 kV). The average particle size was measured using ImageJ software (v.1.8.0).

### 4.7. Fourier Transform Infrared (FTIR) Analysis

The functional groups of the isolated polysaccharide and commercially available fucoidan were characterized through FTIR analysis using the KBr technique (Bruker FTIR, Alpha II spectrometer, Bruker, Leipzig, Germany). In order to prepare analytical samples, potassium bromide (KBr) was used. The sample combination was then compressed into a pellet shape resembling a disk and applied to an FTIR spectrometer. FTIR spectra were observed between 500 and 3500 cm^−1^. The spectrum of each sample was recorded, examined, and compared with that of commercially available fucoidan [27].

### 4.8. Determination of PGE2 and Pro-Inflammatory Cytokine Production

RAW 264.7 cells were pretreated with F5 (50, 100, and 200 µg/mL) prior to LPS stimulation (1 g/mL). The expression levels of PGE2 and pro-inflammatory cytokines (TNF-α, IL-1β, and IL-6) in the culture medium were assessed using commercial ELISA kits in accordance with the manufacturer’s instructions [28].

### 4.9. Western Blot Analysis

Western blot analysis was conducted to ascertain the impact of F5 on the protein expression levels of iNOS, COX-2, NF-κB, and MAPK in the LPS-stimulated RAW 264.7 cells. Cells were seeded in a six-well plate at a density of 1 × 10^5^ cells/mL and incubated for 24 h before exposure to F5. The cells were then stimulated with LPS. Following cell harvesting, proteins were extracted, measured using a BCA protein assay kit, and then separated using gel electrophoresis on 12% sodium dodecyl sulfate–polyacrylamide gels. Blotting was performed on nitrocellulose membranes. The membranes were first blocked with 5% skim milk, following primary and secondary antibodies. The signals were developed using a chemiluminescent substrate (Cyanagen Srl, Bologna, Italy) and photographed using a FUSION SOLO Vilber Lourmat system (Paris, France). The intensity of the resulting bands was measured using ImageJ software.

### 4.10. In Vivo Evaluation of F5 Anti-Inflammatory Activity in a Zebrafish Model

#### 4.10.1. Zebrafish Maintenance

Adult zebrafish were obtained, cared for, and bred by a commercial supplier (Seoul Aquarium, Seoul, South Korea). They received two daily feedings of commercial fish food. Embryos that underwent fertilization were used for further experiments. This experiment was conducted in accordance with the experimental animal guidelines provided by the Jeju National University Animal Center and approved by the Animal Care and Use Committee of Jeju National University (protocol 2020-0049).

#### 4.10.2. Survival Rate and Heart Rate Analysis in LPS-induced Zebrafish

At 7–9 h post-fertilization (hpf), the zebrafish embryos were transferred to a 12-well plate with embryo medium (15 embryos/well). The embryos were treated with F5 (50, 100, and 200 µg/mL) and incubated for 1 h. Embryos were then stimulated with 10 µg/mL of LPS. The survival rate of the embryos was measured, and hatched embryos were continuously maintained. The heart rate of the stimulated zebrafish was recorded 2 days post-fertilization (dpf) [38].

#### 4.10.3. Cell Death, ROS, and NO Production Analysis

Larvae at 3 dpf were used for the analysis of LPS-induced cell death, ROS, and NO production using acridine orange staining (7 µg/mL), Diaminofluorophore,4-amino-5-methylamino-2,7-difluorofluorescein diacetate (DA-FMDA), and DCFH-DA, respectively [19].

### 4.11. Statistical Analysis

GraphPad Prism 9 statistical analysis software (GraphPad Software Inc. San Diego, CA, USA) was used to assess all data using the mean and standard deviation of the three replicates. Mean values were compared using one-way analysis of variance. Duncan’s multiple range test was used to determine mean separation.

## 5. Conclusions

In conclusion, using anion exchange chromatography and celluclast-assisted enzymatic extraction, we extracted fucoidan from *P. arborescens*. According to the compositional analysis data, F5 had the highest sulfate and fucose concentrations. Results from SEM imaging revealed that F5 had distinct morphological traits but lower particle sizes than PAC and PACC. Through the concentration-dependent reduction of NO, PGE2, and pro-inflammatory cytokines (such as TNF-α, IL-6, and IL-1β), F5 prevented LPS-induced inflammation in RAW 264.7 cells. The protective action was linked to the downregulation of iNOS, COX-2, MAPK, and the NF-κB signaling pathway, according to mechanistic studies. In addition, the addition of F5 decreased LPS-induced NO production, ROS production, and cell death in zebrafish. Consequently, F5 reduced LPS-induced inflammation and may be used as a general anti-inflammatory treatment. However, the use of *P. arborescens* in the functional food sector requires further research.

## Figures and Tables

**Figure 1 marinedrugs-22-00109-f001:**
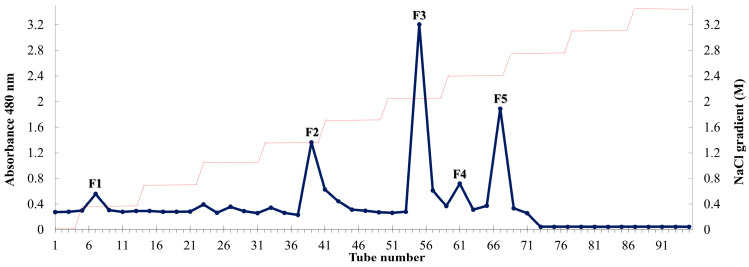
Purification of fucoidan fractions from *P. arborescens* via DEAE-C. Chromatogram of anion-exchange chromatography of PACC.

**Figure 2 marinedrugs-22-00109-f002:**
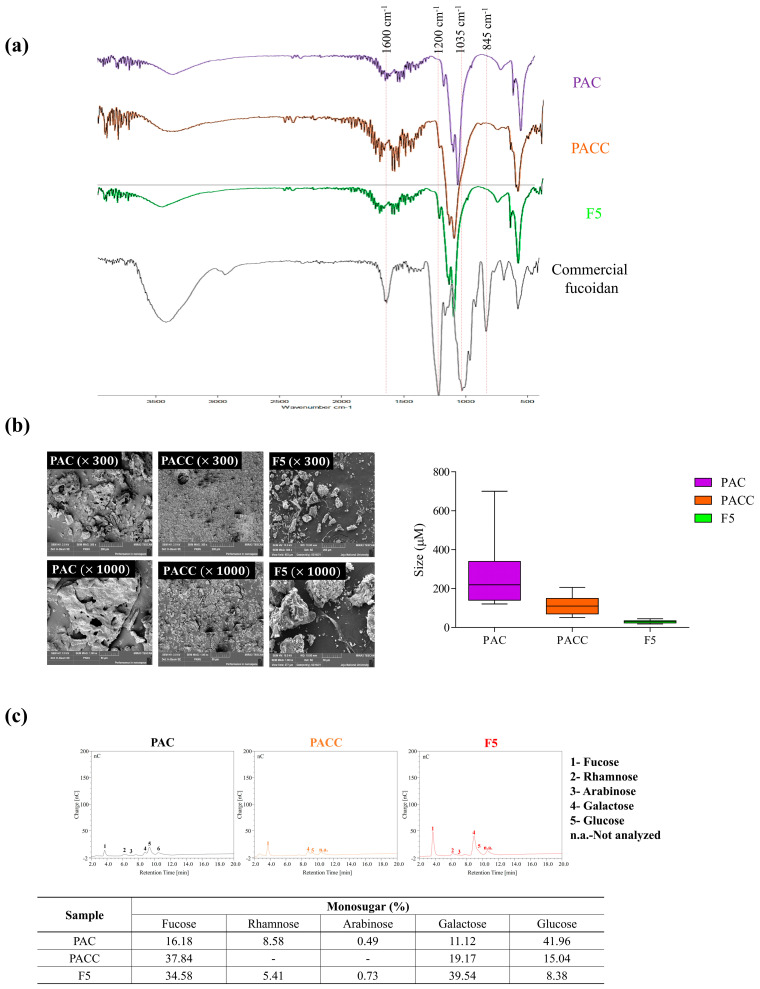
Characterization of structural features and monosaccharide compositions of PAC, PACC, and F5. (**a**) FTIR analysis, (**b**) SEM image analysis and particle size, and (**c**) monosaccharide composition analysis.

**Figure 3 marinedrugs-22-00109-f003:**
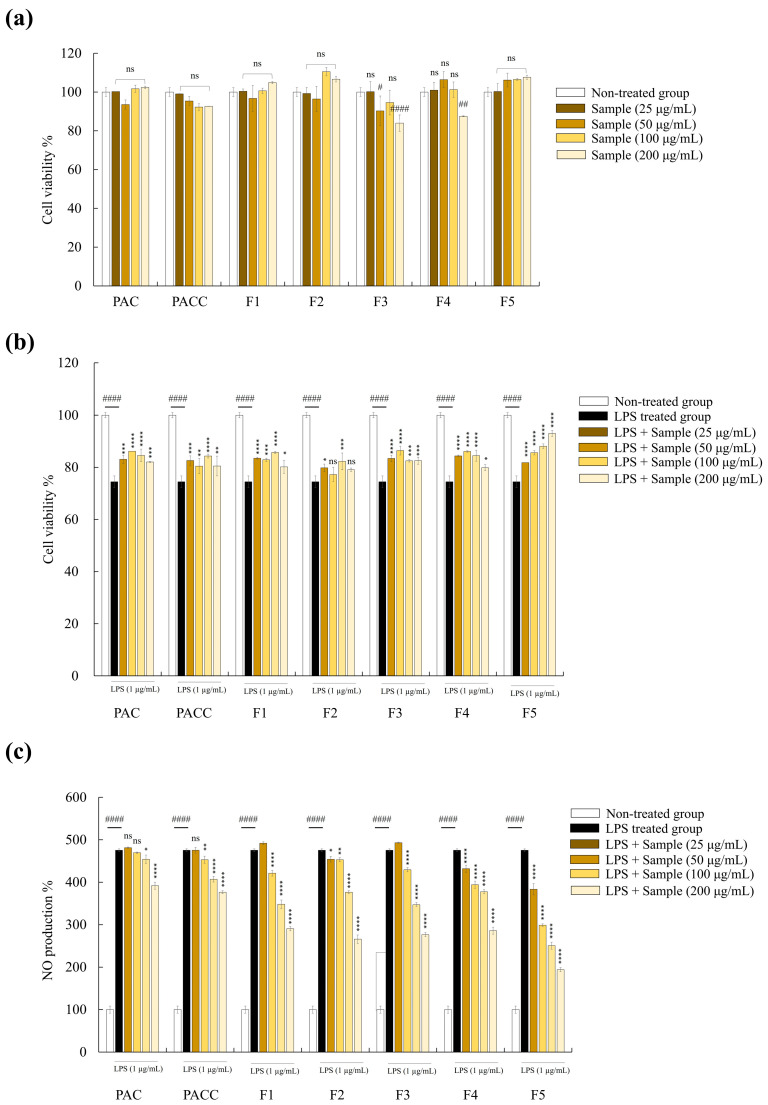
Anti-inflammatory potential of fucoidan fractions (F1–F5) in LPS-induced RAW 264.7 cells. Toxicity (**a**), cell viability (**b**), and NO production (**c**) in LPS-induced RAW 264.7 cells. Values are significantly different from the LPS-treated group (* *p* < 0.05, ** *p* < 0.01, *** *p* < 0.005, and **** *p* < 0.0001) and from the control group (^#^
*p* < 0.05, ^##^
*p* < 0.01, ^####^
*p* < 0.0001). Significance representation was ns, not significant.

**Figure 4 marinedrugs-22-00109-f004:**
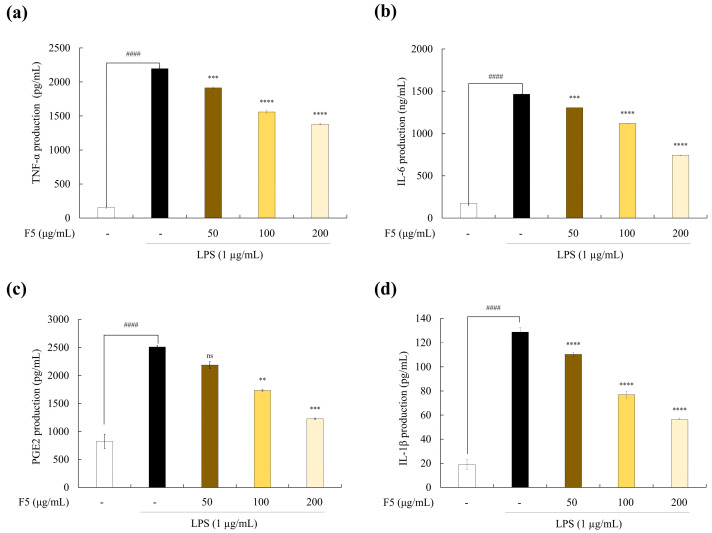
Effect of F5 on the inflammatory regulators. (**a**) TNF-α, (**b**) IL-6, (**c**) PGE-2, and (**d**) IL-1β. Experiments were performed in triplicate, and the results are represented as means ± SD. Values are significantly different from the LPS-treated group (** *p* < 0.01, *** *p* < 0.005, and **** *p* < 0.0001) and from the control group (^####^
*p* < 0.0001). Significance representation was ns, not significant.

**Figure 5 marinedrugs-22-00109-f005:**
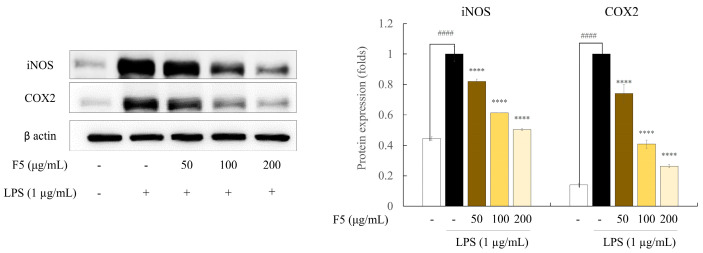
Effect of F5 on iNOS and COX-2 protein expression in LPS-induced Raw 264.7 cells. Values are significantly different from the LPS-treated group (**** *p* < 0.0001) and from the control group (^####^
*p* < 0.0001).

**Figure 6 marinedrugs-22-00109-f006:**
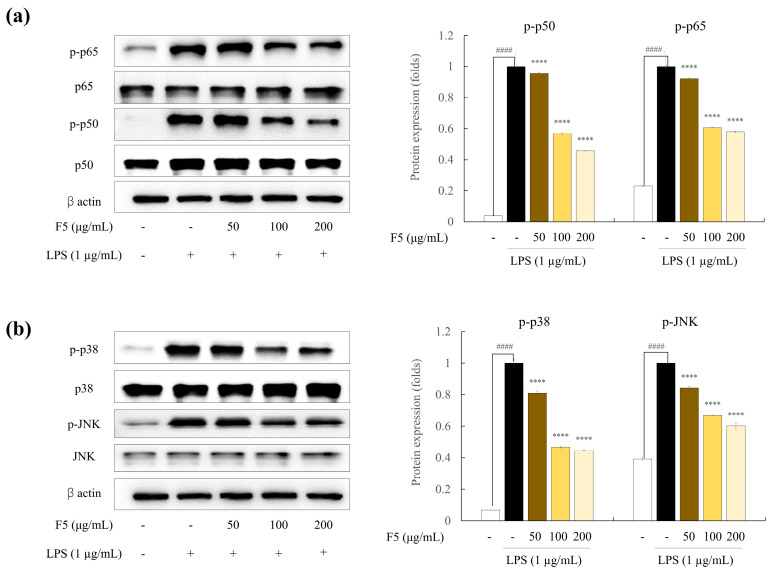
Inhibitory activity of F5 against inflammatory pathway protein expression. (**a**) NF-κB pathway protein expression and (**b**) MAPK pathway protein expression. Values are significantly different from the LPS-treated group (**** *p* < 0.0001) and from the control group (^####^
*p* < 0.0001).

**Figure 7 marinedrugs-22-00109-f007:**
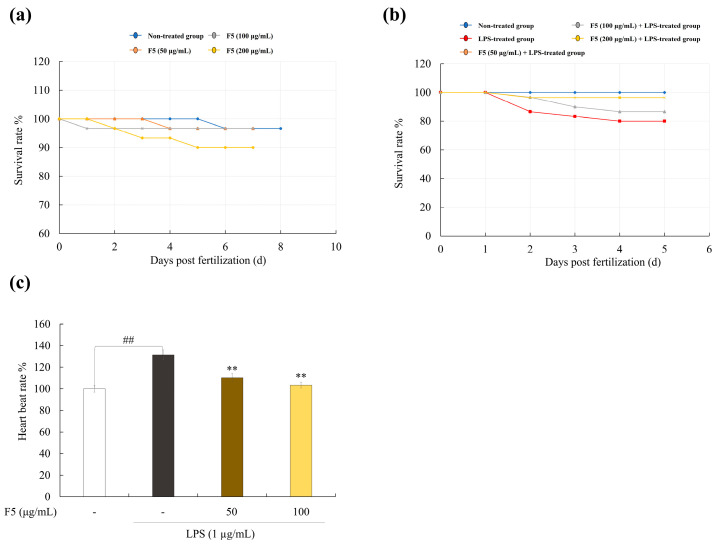
Effect of F5 on LPS-induced survival rate and heartbeat rate alternation in zebrafish. (**a**) Sample toxicity on zebrafish embryo, (**b**) protective effect of F5 on LPS-induced survival rate, and (**c**) heartbeat rate in LPS-induced zebrafish. Experiments were performed in triplicate, and the results are represented as means ± SD. Values are significantly different from the LPS-treated group (** *p* < 0.01) and from the control group (^##^
*p* < 0.01).

**Figure 8 marinedrugs-22-00109-f008:**
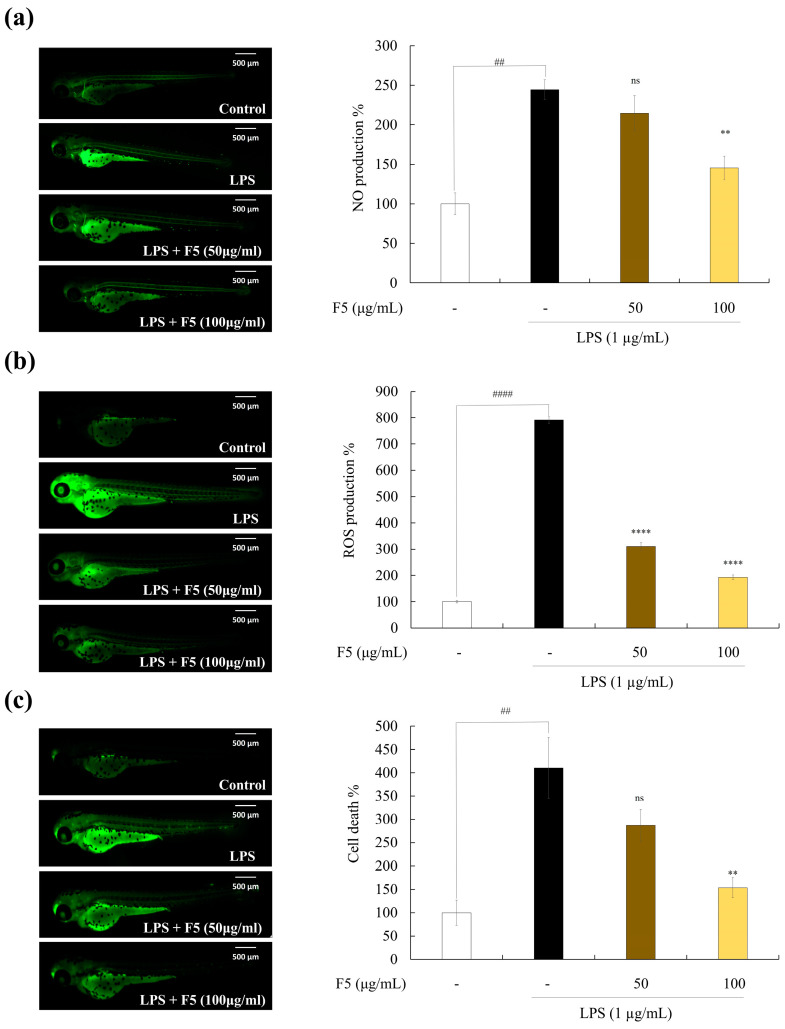
Protective effect of F5 against LPS-induced damage in zebrafish. (**a**) NO production inhibition, (**b**) ROS production inhibition, and (**c**) LPS-induced cell death inhibition. Experiments were performed in triplicate, and the results are represented as means ± SD. Values are significantly different from the LPS-treated group (** *p* < 0.01 and **** *p* < 0.0001) and from the control group (^##^
*p* < 0.01 and ^####^
*p* < 0.0001).

**Table 1 marinedrugs-22-00109-t001:** Proximate compositions of PAC, PACC, and fucoidan fractions.

	Proximate Compositions (%)			
Samples	Polysaccharide	Protein	Polyphenol	Sulfate
PAC	38.93 ± 0.31	13.33 ± 1.62	11.50 ± 0.61	5.44 ± 0.54
PACC	74.80 ± 0.43	10.51 ± 1.22	7.71 ± 0.04	3.31 ±1.14
F1	74.55 ± 1.01	4.01 ± 0.04	3.33 ± 1.82	11.39 ± 1.00
F2	71.66 ± 0.56	3.70 ± 0.62	2,51 ± 0.55	16.77 ± 1.81
F3	65.71 ± 0.94	2.04 ± 0.48	2.01 ±1.00	23.44 ± 0.95
F4	58.44 ± 1.21	1.98 ± 0.82	2.51 ± 1.11	29.48 ± 1.12
F5	51.54 ± 0.33	1.44 ± 0.96	1.92 ± 1.03	38.74 ± 1.05

## Data Availability

Data is contained within the article.

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
