# Peer review of "Investigation of Physical Characteristics and In Vitro Anti-Inflammatory Effects of Fucoidan from Padina arborescens: A Comprehensive Assessment against Lipopolysaccharide-Induced Inflammation"

_marinedrugs, 2024, doi:10.3390/md22030109_

Round 1

Reviewer 1 Report

Comments and Suggestions for Authors

I have read the manuscript and have several questions and recommendations.
1. The authors write that the algae was collected in March 2019. When was the work described in this manuscript done? How long has the seaweed been stored? Almost 5 years???? Numerous experiments have shown that algae metabolites are subject to decomposition during storage, and quite quickly (see, for example, https://doi.org/10.1007/s10811-020-02225-x and others). Please explain and provide data confirming that seaweed can be stored for such a long time.
2. In section 4.2 it is necessary to indicate who identified the algae, where is the herbarium specimen stored?
3. Please provide a literary reference for the technology of fucoidan isolation and purification.
4. What was the activity of the enzyme?
5. Please indicate the yield of crude, purified fucoidan and all fractions. Include this information in Table 1.
6. Please indicate the origin of the commercially available fucoidan to which you are comparing your samples.
7. Please indicate literary references for the phrase “Compositional analysis was performed according to the official methods of AOAC International”.
8. In sections 2.3 and 2.4, provide data for all fractions.
9. For sections 2.5-2.10, an additional positive control group must be included. As a positive control, I recommend using a medicine widely used to treat inflammation, for example, diclofenac or similar.
10. It was previously shown that Fucoidan from the brown alga Fucus vesiculosus significantly inhibits the cyclooxygenase-2 (COX-2) enzyme (IC50 4.3 μg mL−1) with a greater selectivity index (lg(IC80 COX-2/IC80COX-1) , −1.55) than the synthetic non-steroidal anti-inflammatory drug indomethacin (lg(IC80 COX-2/IC80COX-1), −0.09). Fucoidan attenuated the lipopolysaccharide-induced expression of mitogen-activated protein kinase p38. Compare your data with previously obtained data. Can the composition of fucoidan affect the activity?
11. The anti-inflammatory activity of fucoidan is known to depend not only on the type of algae, but also on the composition. It has previously been shown that the anti-inflammatory effects of fucoidan could be mediated via the inhibition of protein denaturation. The inhibition was concentration-dependent and strongly correlated with the fucose content and moderate with sulfate content. Please comment on whether there is a connection between the activity of your fucoidan and its fractions containing polyphenols and/or sulfates. Provide statistics to support your conclusions.
12. The names of algae should be in italics.
13. The phrase in the Conclusion “The present study findings are in line with other investigations of fucoidan from Sargassum fusiforme and Codium fragile [38-40]” does not allow us to conclude that the work is completed.

Author Response

Marine drugs

Manuscript reference no: marinedrugs-2865498

Title: Investigation of physical characteristics and in vitro anti-inflammatory effects on fucoidan from Padina arborescens: A comprehensive assessment against LPS-induced inflammation

Dear Editor

We appreciate the time and effort taken by you for reviewing our manuscript. We are delighted to revise our manuscript so that it may be of sufficient quality for publication in your esteemed journal. The editors have given valuable insights that have improved our manuscript. We have done our best to respond to the editors’ comments and have modified our manuscript accordingly. It is our most sincere hope that this revised manuscript would meet the journal requirements for publication.

Reviewer 1#

Comment 01: The authors write that the algae was collected in March 2019. When was the work described in this manuscript done? How long has the seaweed been stored? Almost 5 years???? Numerous experiments have shown that algae metabolites are subject to decomposition during storage, and quite quickly (see, for example, https://doi.org/10.1007/s10811-020-02225-x and others). Please explain and provide data confirming that seaweed can be stored for such a long time.

Response: Thank you for your comment. Padina boryana collected in March 2019 and it directly washed, dried, and directly extracted. So, we can exclude the decomposition on the Padina boryana and it’s extract.

Comment 02: In section 4.2 it is necessary to indicate who identified the algae, where is the herbarium specimen stored?

Response: Thank you for your valuable comment. This specimen stored at formalin solution after collection and the phylogenetic identification performed by expert in marine plant physiology and ecology.

Comment 03: Please provide a literary reference for the technology of fucoidan isolation and purification.

Response: Fucoidan isolation and purification performed by DEAE, ion-exchange chromatography. The related references are provided below.

References

Antioxidant potential of low molecular weight fucoidans from Sargassum autumnale against H2O2-induced oxidative stress in vitro and in zebrafish models based on molecular weight changes, 2022, Food chemistry, Volume 384

Structural characterization and anti-inflammatory potential of sulfated polysaccharides from Scytosiphon lomentaria; attenuate inflammatory signaling pathways, 2023, Journal of Functional Foods, Volume 102

Comment 04: What was the activity of the enzyme?

Response: Yes, this study we measured extraction yield. Celluclast shows 26% extraction yield and deionized water extraction shows 9% extraction yield. These results indicated that the celluclast activity does not disappear. In addition, we added the extraction yield of deionized water extract in the manuscript.

Comment 05: Please indicate the yield of crude, purified fucoidan and all fractions. Include this information in Table 1.

Response: Thank you for your comment. This study we didn’t measured the extraction yields of crude and purified fucoidans. In my knowledge, PACC show 2-3 % yield and purified fucoidan show below 1% of yield.

Comment 06: Please indicate the origin of the commercially available fucoidan to which you are comparing your samples.

Response: Commercial fucoidan purchased from SIGMA. Fucoidan from Undaria pinnatifida (F8315) used for FTIR analysis.

Comment 07: Please indicate literary references for the phrase “Compositional analysis was performed according to the official methods of AOAC International”.

Response: Thank you for the comment. We added official AOAC international method in the manuscript.

Comment 08: In sections 2.3 and 2.4, provide data for all fractions.

Response: Thank you for your comment. Section 2.3 and Section 2.4 we shows the changes of morphology and monosaccharide compositions between PAC, PACC and F5. Normally, morphology of purified fucoidan fraction showed similar patterns and monosaccharide composition especially Fucose content highly increased in last DEAE fraction. Actually, the absolute value on Fucose increased in F5(0.047 mg/mg) compared with PAC (0.014 mg/mg) and PACC (0.021 mg/mg).

Comment 09: For sections 2.5-2.10, an additional positive control group must be included. As a positive control, I recommend using a medicine widely used to treat inflammation, for example, diclofenac or similar.

Response: Thank you for the comment. Current study, we didn’t used positive control. However when we start follow up study we will use positive control such as diclofenac or dexamethasone.

Comment 10: It was previously shown that Fucoidan from the brown alga Fucus vesiculosus significantly inhibits the cyclooxygenase-2 (COX-2) enzyme (IC50 4.3 μg mL−1) with a greater selectivity index (lg(IC80 COX-2/IC80COX-1) , −1.55) than the synthetic non-steroidal anti-inflammatory drug indomethacin (lg(IC80 COX-2/IC80COX-1), −0.09). Fucoidan attenuated the lipopolysaccharide-induced expression of mitogen-activated protein kinase p38. Compare your data with previously obtained data. Can the composition of fucoidan affect the activity?

Response: Thank you for your valuable comment. Actually, the sulfate highly affected the biologic activity. In our findings, we found that F5 showed most high sulfate compared with PAC, PACC and another fucoidan fractions.

Comment 11: The anti-inflammatory activity of fucoidan is known to depend not only on the type of algae, but also on the composition. It has previously been shown that the anti-inflammatory effects of fucoidan could be mediated via the inhibition of protein denaturation. The inhibition was concentration-dependent and strongly correlated with the fucose content and moderate with sulfate content. Please comment on whether there is a connection between the activity of your fucoidan and its fractions containing polyphenols and/or sulfates. Provide statistics to support your conclusions.

Response: Thank you for the comment. Our findings also showed that F5 contained high sulfate and fucose levels and it affected their biologic activity. In vitro study, F5 showed strong NO inhibitory effect on LPS induced RAW 264.7 cells and it highly attenuated the pro-inflammatory cytokines and PGE2. WB blots showed that F5 remarkably reduced Inos-COX2 and NF-Κb and MAPK related protein expressions.

Comment 12: The names of algae should be in italics.

Response: Thank you for highlighting this mistake. We changed all the algae name in italics.

Comment 13: The phrase in the Conclusion “The present study findings are in line with other investigations of fucoidan from Sargassum fusiforme and Codium fragile [38-40]” does not allow us to conclude that the work is completed.

Response: We agreed on your comment. We removed unnecessary sentence in manuscript.

Reviewer 2 Report

Comments and Suggestions for Authors

Fucoidan, isolated and purified from the hydrolysate of Padina arborescens (PAC), was used to inhibit LPS-induced inflammation in RAW 264.7 cells, and its anti-inflammatory potential was evaluated. From the perspective of work content, the authors have moved from material characterization, to exploration of anti-inflammatory mechanisms at the cellular level, and then to animal experiments. This is a relatively complete work system, but also with novelty. However, there are still some problems. If these problems can be solved, I may consider publishing this article.

1. The author has done a lot of work on the structure of polysaccharides, but in the discussion section, this part of the content can not be seen to participate in the discussion, and the discussion is seriously inadequate. If the length is long, it is suggested to set up sub-headings for discussion.

2. The Conclusions section is suggested to be supplemented.

3. The ordinate of Figure 3C does not correspond to the Figure. Is Figure 3C the content of NO production?

4. The structural characterization part involves NMR, but the relevant data are not presented.

5. The language suggestions of the article should be polished by native English speakers.

6. The references from our own research group are too high in the article, so it must be revised.

Comments on the Quality of English Language

The language suggestions of the article should be polished by native English speakers.

Author Response

Marine drugs

Manuscript reference no: marinedrugs-2865498

Title: Investigation of physical characteristics and in vitro anti-inflammatory effects on fucoidan from Padina arborescens: A comprehensive assessment against LPS-induced inflammation

Dear Editor

We appreciate the time and effort taken by you for reviewing our manuscript. We are delighted to revise our manuscript so that it may be of sufficient quality for publication in your esteemed journal. The editors have given valuable insights that have improved our manuscript. We have done our best to respond to the editors’ comments and have modified our manuscript accordingly. It is our most sincere hope that this revised manuscript would meet the journal requirements for publication.

Reviewer 2#

Fucoidan, isolated and purified from the hydrolysate of Padina arborescens (PAC), was used to inhibit LPS-induced inflammation in RAW 264.7 cells, and its anti-inflammatory potential was evaluated. From the perspective of work content, the authors have moved from material characterization, to exploration of anti-inflammatory mechanisms at the cellular level, and then to animal experiments. This is a relatively complete work system, but also with novelty. However, there are still some problems. If these problems can be solved, I may consider publishing this article.

Comment 01: The author has done a lot of work on the structure of polysaccharides, but in the discussion section, this part of the content can not be seen to participate in the discussion, and the discussion is seriously inadequate. If the length is long, it is suggested to set up sub-headings for discussion.

Response: Thank you for the comment. Current study we isolate and purified fucoidan from P. arborescens and we added in manuscript.

Comment 02: The Conclusions section is suggested to be supplemented.

Response: Thank you for highlighting our mistake. We added conclusion in manuscript.

Comment 03: The ordinate of Figure 3C does not correspond to the Figure. Is Figure 3C the content of NO production?

Response: Thank you for highlighting our mistake. We thoroughly confirmed thin in the whole manuscript.

Comment 04: The structural characterization part involves NMR, but the relevant data are not presented.

Response: I apologize for the confusion. We have removed unnecessary sections from the paper.

Comment 05: The language suggestions of the article should be polished by native English speakers.

Response: We agreed on your opinion. This manuscript edited by EDITAGE, English correction company. We attached the certificate below.

Comment 06: The references from our own research group are too high in the article, so it must be revised.

Response: Yes, we deleted unnecessary reference from the manuscript.

Reviewer 3 Report

Comments and Suggestions for Authors

This manuscript describes a study of the anti-inflammatory properties of a fucose-rich polysaccharide preparation from the brown seaweed Padina arborescens. The fucoidan fraction F5 was found to protect RAW 264.7 cells from LPS-induced inflammatory activities such as NO production, and cytokine generation, through regulation of iNOS/COX2, NF-kB and MAPK signalling pathways. In addition, the authors provide some structural characterisation, though this is incomplete and unsatisfactory (see below). The manuscript is not always well written, and in some places it is not possible to understand. The text needs to be rewritten by someone with expertise in scientific writing in English. There are also numerous errors in figure numbering and the figures are not well presented.

Most concerning of all is the FT-IR data and interpretation, presented in Figure 2a and the text. The spectrum of a commercial fucoidan sample is used as a reference material, but no information is given as to the source of this sample, including its purity and the species of origin. Comparison between the P. arborescens fucoidan spectra and the commercial sample shows that the bands at 1200 cm-1 and 845 cm-1 in the reference spectrum do not coincide with signals in the F5 spectrum. The text is optimistic in its interpretation and may mislead any reader who does not look carefully at the figure. Commercial samples of fucoidan are usually impure and may not have the same structure as fucoidan from P. arborescens, and so may be unsuitable for reference use. It is recommended to remove the FT-IR analysis from this manuscript. NMR spectroscopy (a more informative but more complex technique) is mentioned in the Abstract (line 23) but not included in the paper.

Minor comments:

Figures 2a, 2c, 7a and 7b are to small to be legible

Figures 3b and 3c: the bar chart keys do not explain the white and black bar colours. Are they -ve and +ve controls?

Figure 3c: the vertical axis label is ‘cell viability%’ which looks wrong if the data presented is NO production.

Panels in Fig. 5 are not labelled a and b. The caption of Fig. 5 mentions panels b and c which are not present.

Lines 50, 53, and other locations: what is the meaning of the word ‘vivid’ in this context?

Line 149: ‘Fig 2’ is wrong (maybe Fig. 4?)

Line 158: ‘Fig. 4a’ is wrong (maybe 5a?)

Lines 173 and 180: Fig. 4a and 4b are wrong (maybe Fig. 6?)

Line 190: Fig. 5 is wrong (Fig. 7?)

Line 205: Fig. 6 is wrong (maybe Fig. 8?)

Lines 216-217: This sentence ‘Marine organisms renowned immerse source of secondary metabolites which showed vivid biological potentials’ is completely incomprehensible. What is the relevance of secondary metabolites in the context of fucoidans? Please delete this; also please delete the first sentence in the Abstract (line 18).

Lines 243-249: This FT-IR passage is not based in the data presented in Fig. 2a.

Line 259: ‘heparin sulfate’ should read ‘heparan sulfate’.

Comments on the Quality of English Language

The quality of English in this manuscript varies between acceptable and poor. The poor sections need to be rewritten either by a colleague with good English skills or by a professional editing service.

Author Response

Marine drugs

Manuscript reference no: marinedrugs-2865498

Title: Investigation of physical characteristics and in vitro anti-inflammatory effects on fucoidan from Padina arborescens: A comprehensive assessment against LPS-induced inflammation

Dear Editor

We appreciate the time and effort taken by you for reviewing our manuscript. We are delighted to revise our manuscript so that it may be of sufficient quality for publication in your esteemed journal. The editors have given valuable insights that have improved our manuscript. We have done our best to respond to the editors’ comments and have modified our manuscript accordingly. It is our most sincere hope that this revised manuscript would meet the journal requirements for publication.

Reviewer 3#

This manuscript describes a study of the anti-inflammatory properties of a fucose-rich polysaccharide preparation from the brown seaweed Padina arborescens. The fucoidan fraction F5 was found to protect RAW 264.7 cells from LPS-induced inflammatory activities such as NO production, and cytokine generation, through regulation of iNOS/COX2, NF-kB and MAPK signalling pathways. In addition, the authors provide some structural characterisation, though this is incomplete and unsatisfactory (see below). The manuscript is not always well written, and in some places it is not possible to understand. The text needs to be rewritten by someone with expertise in scientific writing in English. There are also numerous errors in figure numbering and the figures are not well presented.

Most concerning of all is the FT-IR data and interpretation, presented in Figure 2a and the text. The spectrum of a commercial fucoidan sample is used as a reference material, but no information is given as to the source of this sample, including its purity and the species of origin. Comparison between the P. arborescens fucoidan spectra and the commercial sample shows that the bands at 1200 cm-1 and 845 cm-1 in the reference spectrum do not coincide with signals in the F5 spectrum. The text is optimistic in its interpretation and may mislead any reader who does not look carefully at the figure. Commercial samples of fucoidan are usually impure and may not have the same structure as fucoidan from P. arborescens, and so may be unsuitable for reference use. It is recommended to remove the FT-IR analysis from this manuscript. NMR spectroscopy (a more informative but more complex technique) is mentioned in the Abstract (line 23) but not included in the paper.

Comment 01: Figures 2a, 2c, 7a and 7b are to small to be legible

Response: Thank you for the comment. We increased the size of Figures 2a, 2c, 7a and 7b in manuscript.

Comment 02: Figures 3b and 3c: the bar chart keys do not explain the white and black bar colours. Are they -ve and +ve controls?

Response: Thank you for the comment. We changed -ye and +ye control in Figure 3.

Comment 03: Figure 3c: the vertical axis label is ‘cell viability%’ which looks wrong if the data presented is NO production.

Response: Thank you for highlighting this mistake. We have changed this in manuscript.

Comment 04: Panels in Fig. 5 are not labelled a and b. The caption of Fig. 5 mentions panels b and c which are not present.

Response: I apologize for the confusion. We revised Figure 5 in manuscript. Figure 5 shows the iNOS and COX-2 expression levels in LPS induced Raw 264.7 cells.

Comment 05: Lines 50, 53, and other locations: what is the meaning of the word ‘vivid’ in this context?

Response: Thank you for the comment. We changed it in manuscript.

Comment 06: Line 149: ‘Fig 2’ is wrong (maybe Fig. 4?)

Response: Sorry for the confusion. We revised this in manuscript.

Comment 07: Line 158: ‘Fig. 4a’ is wrong (maybe 5a?)

Response: Sorry for the confusion. We revised this in manuscript.

Comment 08: Lines 173 and 180: Fig. 4a and 4b are wrong (maybe Fig. 6?)

Response: Sorry for the confusion. We revised this in manuscript.

Comment 09: Line 190: Fig. 5 is wrong (Fig. 7?)

Response: Sorry for the confusion. We revised this in manuscript.

Comment 10: Line 205: Fig. 6 is wrong (maybe Fig. 8?)

Response: Sorry for the confusion. We revised this in manuscript.

Comment 11: Lines 216-217: This sentence ‘Marine organisms renowned immerse source of secondary metabolites which showed vivid biological potentials’ is completely incomprehensible. What is the relevance of secondary metabolites in the context of fucoidans? Please delete this; also please delete the first sentence in the Abstract (line 18).

Response: We agreed on your opinion. We deleted unnecessary sentence in the manuscript.

Comment 12: Lines 243-249: This FT-IR passage is not based in the data presented in Fig. 2a.

Response: Thank you for the comment. We revised it in manuscript.

Comment 13: Line 259: ‘heparin sulfate’ should read ‘heparan sulfate’.

Response: Thank you for the comment. We revised it in manuscript.

Round 2

Reviewer 1 Report

Comments and Suggestions for Authors

I have read the revision manuscript and the authors' responses and comments.
Despite significant corrections to the manuscript, I still have fundamental questions.
1. Numerous published articles report instability of seaweed components, including frozen, dried, etc. (see for example https://doi.org/10.1007/s10811-020-02225-x and others).

Authors must provide analysis data from a fresh batch of algae to support their findings.
2. In section 4.2 it is necessary to indicate who identified the algae, where is the herbarium specimen stored?
3. Please check the yields on the new batch of raw materials and provide information. This is important information that will allow you to draw a conclusion about the value of your work.
4. For sections 2.5-2.10, an additional positive control group must be included. As a positive control, I recommend using a medicine widely used to treat inflammation, for example, diclofenac or similar. If the authors cannot provide relevant information in comparison with the comparator drug, this section should be deleted or the manuscript should be submitted to another journal that does not focus on marine medicines.
5. It was previously shown that Fucoidan from the brown alga Fucus vesiculosus significantly inhibits the cyclooxygenase-2 (COX-2) enzyme (IC50 4.3 μg mL−1) with a greater selectivity index (lg(IC80 COX-2/IC80COX-1 ) , −1.55) than the synthetic non-steroidal anti-inflammatory drug indomethacin (lg(IC80 COX-2/IC80COX-1), −0.09). Fucoidan attenuated the lipopolysaccharide-induced expression of mitogen-activated protein kinase p38. Compare your data with previously obtained data. Can the composition of fucoidan affect the activity?
6. The anti-inflammatory activity of fucoidan is known to depend not only on the type of algae, but also on the composition. It has previously been shown that the anti-inflammatory effects of fucoidan could be mediated via the inhibition of protein denaturation. The inhibition was concentration-dependent and strongly correlated with the fucose content and moderate with sulfate content. Please comment on whether there is a connection between the activity of your fucoidan and its fractions containing polyphenols and/or sulfates. Provide statistics to support your conclusions.
As presented, I cannot recommend the manuscript for publication in the journal Marine Drugs.

Author Response

Reviewer 1#

Commnet: Numerous published articles report instability of seaweed components, including frozen, dried, etc. (see for example https://doi.org/10.1007/s10811-020-02225-x and others).

Respond: Thank you for your valuable comment. We can exclude some instability effects. After collecting, Padina arborescence directly dried and extracted and stored in -20 freezer before use.

Commnet: Authors must provide analysis data from a fresh batch of algae to support their findings.

Respond: Thank you for the comment. We collected alive Padina arborescence from the shore of Jeju Isand in Korea. This was directly collected from nature, and no separate arrangement has been made

Commnet: In section 4.2 it is necessary to indicate who identified the algae, where is the herbarium specimen stored?

Respond: The sample has been identified by an expert in algal ecology, and the specimen is stored at the Marine Life and Biotechnology Laboratory of the Department of Marine Biology, Jeju National University.

Commnet: Please check the yields on the new batch of raw materials and provide information. This is important information that will allow you to draw a conclusion about the value of your work.

Respond: Thank you for the comment. We mentioned extraction yields in manuscript (Line 82)

Commnet: For sections 2.5-2.10, an additional positive control group must be included. As a positive control, I recommend using a medicine widely used to treat inflammation, for example, diclofenac or similar. If the authors cannot provide relevant information in comparison with the comparator drug, this section should be deleted or the manuscript should be submitted to another journal that does not focus on marine medicines.

Respond: Thank you for the comment. Current study, we didn’t used positive control. However when we start follow up study we will use positive control such as diclofenac or dexamethasone.

This paper seems more suitable for submission to the Special Issue on Polysaccharides from the Marine Environment. If possible. kindly consider making the necessary changes for submission to this specific issue. Thank you.

Commnet: It was previously shown that Fucoidan from the brown alga Fucus vesiculosus significantly inhibits the cyclooxygenase-2 (COX-2) enzyme (IC50 4.3 μg mL−1) with a greater selectivity index (lg(IC80 COX-2/IC80COX-1 ) , −1.55) than the synthetic non-steroidal anti-inflammatory drug indomethacin (lg(IC80 COX-2/IC80COX-1), −0.09). Fucoidan attenuated the lipopolysaccharide-induced expression of mitogen-activated protein kinase p38. Compare your data with previously obtained data. Can the composition of fucoidan affect the activity?

Respond: Thank you for the comment. We have incorporated a comparison of our research findings with those of other researchers in the discussion section. (Line 287-289)

Comment: The anti-inflammatory activity of fucoidan is known to depend not only on the type of algae, but also on the composition. It has previously been shown that the anti-inflammatory effects of fucoidan could be mediated via the inhibition of protein denaturation. The inhibition was concentration-dependent and strongly correlated with the fucose content and moderate with sulfate content. Please comment on whether there is a connection between the activity of your fucoidan and its fractions containing polyphenols and/or sulfates. Provide statistics to support your conclusions.

Respond: Thank you for the comment. In Table 1, F5 exhibits the highest content of sulfated groups among the results. Additionally, in the results of NO inhibitory efficacy shown in Figure 3, F5 demonstrates the most superior efficacy. Furthermore, the monosaccharide analysis reveals that while the ratio of fucose to other sugars is similar between PACC and F5, the absolute amount of fucose in F5 is more than twice as high as in PAC and PACC when measured. The raw data for absolute quantities are attached below.

Reviewer 2 Report

Comments and Suggestions for Authors

1. The repetition rate of the article is high, and the author needs to reduce the repetition rate of the article.

Author Response

Reviewer 2#

Comment: The repetition rate of the article is high, and the author needs to reduce the repetition rate of the article.

Respond: Thank you for your valuable feedback. In order to reduce repetition rate throughout the entire paper, I have thoroughly read and revised the entire manuscript.

Reviewer 3 Report

Comments and Suggestions for Authors

The quality of English in this revised version is very much improved.

There is one small but important change required: in the abstract, please remove the words "nuclear magnetic resonance (NMR)". There are no NMR results reported in this paper. 

Author Response

Reviewer 3#

Comment: There is one small but important change required: in the abstract, please remove the words "nuclear magnetic resonance (NMR)". There are no NMR results reported in this paper. \

Respond: Thank you for emphasizing the errors until the end. I have removed the NMR section from the abstract.